# Protective Responses at the Biochemical and Molecular Level Differ between a *Coffea arabica* L. Hybrid and Its Parental Genotypes to Supra-Optimal Temperatures and Elevated Air [CO_2_]

**DOI:** 10.3390/plants11202702

**Published:** 2022-10-13

**Authors:** Gabriella Vinci, Isabel Marques, Ana P. Rodrigues, Sónia Martins, António E. Leitão, Magda C. Semedo, Maria J. Silva, Fernando C. Lidon, Fábio M. DaMatta, Ana I. Ribeiro-Barros, José C. Ramalho

**Affiliations:** 1Department of Biological, Geological and Environmental Sciences (BiGeA), Alma Mater Studiorum, The University of Bologna, Via Irnerio 42, 40126 Bologna, Italy; 2Plant Stress & Biodiversity Lab, Centro de Estudos Florestais (CEF), Instituto Superior Agronomia (ISA), Universidade de Lisboa (ULisboa), Quinta do Marquês, Av. República, Oeiras, 2784-505 Lisboa, Portugal; 3Plant Stress & Biodiversity Lab, Centro de Estudos Florestais (CEF), Instituto Superior Agronomia (ISA), Universidade de Lisboa (ULisboa), Quinta do Marquês, Av. República, Tapada da Ajuda, 1349-017 Lisboa, Portugal; 4Departamento de Engenharia Química, Instituto Superior de Engenharia de Lisboa, Instituto Politécnico de Lisboa, R. Conselheiro Emídio Navarro 1, 1959-007 Lisboa, Portugal; 5Unidade de Geobiociências, Geoengenharias e Geotecnologias (GeoBioTec), Faculdade de Ciências e Tecnologia (FCT), Universidade NOVA de Lisboa (UNL), Monte de Caparica, 2829-516 Caparica, Portugal; 6Departamento de Biologia Vegetal, Universidade Federal Viçosa (UFV), Viçosa 36570-900, MG, Brazil

**Keywords:** antioxidant system, climate change, coffee, elevated carbon dioxide, heat stress, oxidative stress

## Abstract

Climate changes with global warming associated with rising atmospheric [CO_2_] can strongly impact crop performance, including coffee, which is one of the most world’s traded agricultural commodities. Therefore, it is of utmost importance to understand the mechanisms of heat tolerance and the potential role of elevated air CO_2_ (eCO_2_) in the coffee plant response, particularly regarding the antioxidant and other protective mechanisms, which are crucial for coffee plant acclimation. For that, plants of *Coffea arabica* cv. Geisha 3, cv. Marsellesa and their hybrid (Geisha 3 × Marsellesa) were grown for 2 years at 25/20 °C (day/night), under 400 (ambient CO_2_, aCO_2_) or 700 µL (elevated CO_2_, eCO_2_) CO_2_ L^−1^, and then gradually submitted to a temperature increase up to 42/30 °C, followed by recovery periods of 4 (Rec4) and 14 days (Rec14). Heat (37/28 °C and/or 42/30 °C) was the major driver of the response of the studied protective molecules and associated genes in all genotypes. That was the case for carotenoids (mostly neoxanthin and lutein), but the maximal (*α* + *β*) carotenes pool was found at 37/28 °C only in Marsellesa. All genes (except *VDE*) encoding for antioxidative enzymes (catalase, CAT; superoxide dismutases, CuSODs; ascorbate peroxidases, APX) or other protective proteins (HSP70, ELIP, Chape20, Chape60) were strongly up-regulated at 37/28 °C, and, especially, at 42/30 °C, in all genotypes, but with maximal transcription in Hybrid plants. Accordingly, heat greatly stimulated the activity of APX and CAT (all genotypes) and glutathione reductase (Geisha3, Hybrid) but not of SOD. Notably, CAT activity increased even at 42/30 °C, concomitantly with a strongly declined APX activity. Therefore, increased thermotolerance might arise through the reinforcement of some ROS-scavenging enzymes and other protective molecules (HSP70, ELIP, Chape20, Chape60). Plants showed low responsiveness to single eCO_2_ under unstressed conditions, while heat promoted changes in aCO_2_ plants. Only eCO_2_ Marsellesa plants showed greater contents of lutein, the pool of the xanthophyll cycle components (V + A + Z), and *β*-carotene, compared to aCO_2_ plants at 42/30 °C. This, together with a lower CAT activity, suggests a lower presence of H_2_O_2_, likely also associated with the higher photochemical use of energy under eCO_2_. An incomplete heat stress recovery seemed evident, especially in aCO_2_ plants, as judged by the maintenance of the greater expression of all genes in all genotypes and increased levels of zeaxanthin (Marsellesa and Hybrid) relative to their initial controls. Altogether, heat was the main response driver of the addressed protective molecules and genes, whereas eCO_2_ usually attenuated the heat response and promoted a better recovery. Hybrid plants showed stronger gene expression responses, especially at the highest temperature, when compared to their parental genotypes, but altogether, Marsellesa showed a greater acclimation potential. The reinforcement of antioxidative and other protective molecules are, therefore, useful biomarkers to be included in breeding and selection programs to obtain coffee genotypes to thrive under global warming conditions, thus contributing to improved crop sustainability.

## 1. Introduction

Climate changes associated with global warming are expected to endanger ecosystems and food security [1]. Climate changes are believed to be closely driven by greenhouse gas (GHG) emissions into the atmosphere. Among them, air [CO_2_] already exceeds 400 µL L^−1^ and continues to increase at a rate close to 2 µL CO_2_ L^−1^ per year [2], with estimates pointing to unprecedented values between 730 and 1000 µL L^−1^ by 2100, depending on the measures to control future GHGs emissions. This air [CO_2_] increase is believed to trigger a temperature rise between 1.0–1.8 °C (best scenario) and 3.3–5.7 °C (worst scenario without additional efforts to limit the emissions), as compared to 1850–1900 [3,4]. Climate changes are already affecting the frequency and severity of extreme events, such as heat waves, longer and harsher droughts, unpredictable rainfalls, etc. [4]. This will have severe impacts on agricultural ecosystems, with the consequent decline of crop yields, quality [3], and suitable areas, under increasing pressure for feed and food availability to fulfill the demands of a growing world population [5] that is expected to approach 10,000 million people by 2050 [6,7,8].

High temperature is one of the major abiotic stresses that pose growing and serious challenges to plant growth and development [9]. Heat stress affects several physiological processes [10], e.g., it could alter membrane permeability and fluidity, influencing cellular homeostasis [11,12], and cause denaturation and aggregation of proteins [13], cell damage and ion leakage, interfering with important processes such as respiration and photosynthesis [14]. Indeed, the photosynthetic apparatus is highly sensitive to high temperatures [15], namely at the PSII level [9], associated with the dissociation of the D1 protein, but the electron transport chain (ETC) and the oxygen-evolving complex (OEC), and photosynthetic enzymes, such as RuBisCO activase, could also be inactivated [10,16,17]. Additionally, temperature rise stimulates photorespiration and respiration more than photosynthesis due to decreases in the affinity of RuBisCO for CO_2_ and the solubility of CO_2_, both relative to O_2_, thus reducing the relative rate of carboxylation to oxygenation and C-assimilation [18,19,20].

Unfavorable environmental conditions that inhibit energy use through photochemistry may promote the over-reduction of the ETC, and the accumulation of molecules in the excited state [9,21], such as singlet oxygen (^1^O_2_), the singlet chlorophyll (^1^Chl) and triplet state of chlorophyll (^3^Chl), leading to the production of superoxide radical (O_2_*^−^), hydrogen peroxide (H_2_O_2_) and hydroxyl radical (OH^●−^) [22]. The reactive oxygen species (ROS), mainly produced in chloroplasts, mitochondria, and peroxisomes [23], are harmful oxidants able to cause lipid peroxidation, enzyme inactivation, and degradation of pigments, proteins, and DNA [24] leading ultimately to cell apoptosis [25]. The control of highly reactive molecules of Chl and O_2_ is achieved by promoting energy dissipation mechanisms that prevent its formation (e.g., photoprotective pigments) and the expression of enzymatic and non-enzymatic antioxidants that scavenge the ROS already produced [13,24]. Therefore, the reinforcement of mechanisms dedicated to preventing ROS overproduction and/or its efficient scavenging is usually crucial to plant tolerance to a wide number of environmental constraints. That is also the case in *Coffea* spp., which shows a common antioxidative response to several stresses, such as drought [26,27,28,29], high irradiance [30], cold [24,31,32], and heat [33,34,35,36].

Coffee is one of the most world’s traded commodities and popular beverages, consumed by about one-third of the world’s population [37]. It is estimated that the coffee chain involves ca. 100–125 million people from cultivation to the final product for consumption [37,38], based on the production of ca. 25 million smallholder farmers [39], which represent about 60% of the coffee farms, usually of low income [40]. The world’s annual production has reached ca. 10 million tons in recent years [41], yielded from approximately 80 tropical countries, from South and Central America to Africa and Southwest Asia, extending from a latitude of 20–25° N in Hawaii down to 24° S in Brazil [42]. Among the identified 130 species of the *Coffea* genus [43], two species are responsible for almost all the world’s coffee production: *C. arabica* L. (arabica coffee) and *C. canephora* Pierre ex A. Froehner (robusta coffee) [42].

Coffee crop and yield, especially in *C. arabica*, are strongly influenced by climatic variability, particularly extreme temperatures and water deficit [44,45]. With the increase in global mean temperatures, the coffee industry might have to face serious challenges in the future, with negative consequences for the entire supply chain. Climate changes are expected to cause a reduction in coffee crop yields and in suitable land for coffee growth [46,47,48,49]. Traditionally, the optimal annual mean temperature range was stated as 18–21 °C for arabica cultivars [50]. In this way, it was reported that mean air temperatures above 23 °C could accelerate fruit ripening of arabica cultivars, which can cause bean quality loss, and seasonal high temperatures above 33 °C and dryer seasons can greatly reduce floral initiation and increase the production of abnormal reproductive structures, and flower abortion [51,52,53,54]. However, current Arabica cultivars can grow in marginal regions, such as in the northeast of Brazil, where the mean annual temperature can reach 25 °C [55], and elite cultivars can successfully withstand relatively high temperatures [12,34] to a greater extent than traditionally assumed in classical studies [56]. Furthermore, elevated air [CO_2_] (eCO_2_) was reported to play a key role in heat stress resilience in coffee genotypes, with the potential to offset some of the negative impacts of climate change [33,34,57]. Indeed, recent experiments showed that coffee can tolerate heat stress and maintain photosynthetic performance at temperatures up to 37/30 °C (day/night), especially under eCO_2_, and that under 42/34 °C, photosynthesis is greatly affected, but relevant photosynthetic activity is still maintained only under eCO_2_ [33,34].

In this context, the present work was undertaken to test the ability of new elite arabica coffee genotypes to cope with supra-optimal temperatures in the context of climate change, with a focus on the potential protective mechanisms (especially those associated with photo- and antioxidative action, and how these responses can be (or not) affected by eCO_2_. Biochemical and molecular approaches were used to study the effects of supra-optimal temperature under both ambient air [CO_2_] (aCO_2_) or eCO_2_ in terms of (a) carotenoids concentration, (b) cellular activity of antioxidant enzymes, and (c) expression of genes related to the antioxidant and other protective mechanisms. Furthermore, hybrid vigor (heterosis), a way to explore and increase the acclimation capabilities in crops with the progeny exhibiting traits that outperform their parentals [58], was also envisaged in a hybrid resulting from *C. arabica* Marsellesa × Geisha 3 cross.

## 2. Results

### 2.1. Carotenoids Evaluation

At the control temperature (25/20 °C), the single eCO_2_ exposure in Marsellesa altered the carotenoid composition, showing greater concentrations of several pigments, significantly for neoxanthin (50%), the sum of violaxanthin (V), antheraxanthin (A), and zeaxanthin (Z; V + A + Z, 64%), lutein (54%), total carotenoids (54%), and a tendency to greater values of (*α* + *β*) carotene (53%; Table 1). By contrast, individual carotenoids did not respond significantly to eCO_2_ in Geisha 3 and Hybrid plants.

Under aCO_2_, the single impact of temperature rising to 31/25 °C and then further to 37/28 °C led to a general increase of most carotenoids in the three genotypes, although significantly only in some cases, especially in Geisha 3 and Hybrid plants. That was reflected in maximal increases in total carotenoid concentration in Geisha 3 (76%), Hybrid (43%), both at 31/25 °C, and Marsellesa (40%) at 37/28 °C (the latter was not significant), always when compared to their control values.

In detail, maximal increases of neoxanthin and lutein were found in Geisha 3 at 31/25 °C (50% and 39%, respectively) and in the Hybrid at 37/28 °C (41% and 40%, in the same order). Always at 31/25 °C, Geisha 3 and Hybrid plants also showed maximal contents of *α*-carotene (105% and 78%), *β*-carotene (63% and 28%), and (*α* + *β*) carotene (76% and 43%), with the concomitant increase of (*α*/*β*)-carotene ratio. Notably, by 37/28 °C, both zeaxanthin and DEPS values were not significantly altered in none of the genotypes.

As the temperature reached the maximum value (42/30 °C), the carotenoid concentration declined in aCO_2_ plants from their maxima to values that usually did not differ from their controls at 25/20 °C, except for lutein in Geisha 3 plants that maintained an increased value, or violaxanthin that showed reduced values (all genotypes). Still, DEPS tended to greater values than at 25/20 °C, with rises in Geisha 3 (45%), Marsellesa (315%), and Hybrid leaves (98%). A strong decline was observed in carotenes pools, particularly in *α*-carotene, implicating a clear decline in the (*α*/*β*)-carotene ratio of all genotypes.

The plants grown under eCO_2_ submitted to temperature rise showed notable stability of the studied carotenoids, without significant modifications from 25/20 °C up to 37/28 °C (lutein) or even 42/30 °C (neoxanthin, zeaxanthin, V + A + Z, carotenes, total carotenoid) for the three genotypes. That also implicated the stability of DEPS values, even at 42/30 °C, although the (*α*/*β*)-carotene ratio tended to lower values in Geisha 3 and Marsellesa at that temperature.

Only a few significant differences were depicted between the two [CO_2_] conditions regarding the leaf xanthophyll concentration at maximal temperature. Noteworthy were the observations that, at 42/30 °C, the eCO_2_ Marsellesa plants showed greater concentrations of V + A + Z and, especially, lutein than their aCO_2_ counterparts, but such eCO_2_ superimposition reduced DEPS in all genotypes (significantly only in Hybrid) as compared with aCO_2_ plants. Additionally, at 42/30 °C, the eCO_2_ plants tended to have greater values of *α*-carotene (significant in Hybrid), *β*-carotene, (*α* + *β*) carotene (both significant in Marsellesa), and (*α*/*β*) carotene (significant in Hybrid).

Overall, during the recovery period (at 25/20 °C), most carotenoid concentrations returned to values that did not differ from the initial values, usually regardless of CO_2_ condition and genotype. That was the case from Rec4 onwards of neoxanthin (all genotypes), zeaxanthin (only by Rec14 for Hybrid plants), V + A + Z, *α*-carotene, *β*-carotene, and (*α* + *β*) carotene (except the Hybrid plants under aCO_2_ for the latter). In contrast, lutein and total carotenoids maintained increased levels by Rec4 in all genotypes under aCO_2_, and still by Rec14 in Geisha 3 and Hybrid plants, whereas *α*-carotene had reduced concentrations even by Rec14 in the three genotypes and both [CO_2_]. DEPS showed greater values than the control at 25/20 °C for Marsellesa and Hybrid plants both in Rec4 and Rec14.

### 2.2. Antioxidant Enzyme Activity

At control temperature, the single eCO_2_ exposure altered the activity of antioxidative enzymes, depending on the genotype and enzyme (Figure 1 and Figure 2). In Marsellesa, the potential activities of SOD (Figure 1) and GR (Figure 2) declined 49% and 58%, respectively, relative to their aCO_2_ plants, whereas APX and CAT were not affected. Geisha 3 plants showed an approximately doubled APX activity and ca. half of CAT, without changes in SOD and GR. Notably, all four enzymes were mostly insensitive to eCO_2_ in the Hybrid plants.

Under single 37/28 °C exposure, remarkable activity increases were observed for APX, by 95% (Geisha 3), 368% (Marsellesa) and 71% (Hybrid); for CAT, by 94% (Geisha 3), 153% (Marsellesa) and 122% (Hybrid); and for GR by 125% (Geisha 3) and 196% (Hybrid). Instead, SOD activity tended to decrease significantly only in the Hybrid plants, always as compared with their respective activity values at 25/20 °C and aCO_2_.

With further temperature increase to 42/30 °C, APX activity was severely depressed (Figure 1), whereas CAT showed values even greater (Geisha 3 and Hybrid) or unaltered (Marsellesa) as compared with those at 37/28 °C (Figure 2). At this harshest condition (42/30 °C), SOD activity continued to decline (except in the Hybrid), whereas GR values were mostly maintained in all genotypes when compared to the values under 37/28 °C.

The superimposition of eCO_2_ at 37/28 °C altered, in a few cases, the enzyme activities observed under aCO_2_. A clearer case was observed with CAT, whose activity decreased under eCO_2_ both at 37/28 °C and 42/30 °C in all genotypes. Additionally, APX also declined in Geisha 3 (42/30 °C), Marsellhesa and Hybrid (both at 37/28 °C). In contrast, SOD activity increased in Geisha 3 (by 63% at 42/30 °C), Marsellhesa and Hybrid (by 64% and 35%, respectively, both at 37/28 °C), although not showing differences to aCO_2_ at 42/30 °C for the last two genotypes. Marsellesa showed the lowest values among the three genotypes. Finally, GR activity was unchanged between CO_2_ conditions at 37/28 °C and 42/30 °C for the three genotypes, except for a decline at the harshest temperature in Geisha 3.

After 14 days of recovery (Rec14) at 25/20 °C, the aCO_2_ plants tended to approach the value observed at the beginning of the experiment (also at 25/20 °C). That was the case for all enzymes in Geisha 3 and Marsellesa. In the Hybrid plants, there were reduced activities of SOD and APX and greater values of CAT.

Still, by Rec14, the eCO_2_ plants showed a distinct pattern from their aCO_2_ counterparts. All genotypes showed lower SOD activity under eCO_2_ than at aCO_2_ by the end of the experiment (Figure 1), but for the other enzymes, a genotype-dependent situation was observed. In eCO_2_ plants, Geisha 3 showed a declined APX activity, in opposition to the rise in CAT. Marsellesa kept close APX activities between [CO_2_] conditions and a reduction of CAT under eCO_2_ (but always similar to the initial controls), whereas the Hybrid plants showed greater (130%) and lower (89%) values for APX and CAT, respectively. Finally, GR was mostly irresponsive to [CO_2_] by Rec14 in the studied genotypes, showing values close to those at the beginning of the trial (Figure 2).

### 2.3. Expression of Selected Genes Associated with Protective Roles

The single exposure to eCO_2_ at 25/20 °C did not significantly alter the expression of any of the studied genes, irrespective of genotype, in comparison to their aCO_2_ plant counterparts (Table 2).

Regarding the single temperature exposure, a moderate down-regulation impact was observed only for *VDE2* at 37/28 and/or 42/30 °C, together with a total recovery under Rec14 for the three genotypes compared with the initial expression values at 25/20 °C. For all the other studied genes, considering both those encoding for antioxidative enzymes (CAT, the two CuSODs, and the three APX isoforms) and other protective proteins (HSP70, ELIP, Chape20, Chape60), the high temperature was the major driver of gene expression changes, promoting a strong up-regulation at 37/28 °C and a further rise at 42/30 °C, when the highest expression values were usually attained in the three genotypes. Notably, maximal gene expression was always found in Hybrid plants (aCO_2_, 42/30 °C), except for *CAT* (Geisha 3) and *APX_Cyt_* (Marsellesa).

In general, the abundance of gene transcripts associated with antioxidative enzymes was the highest up-regulated in the three genotypes. For instance, the expression of genes encoding for Cu, Zn-SOD (*CuSOD1*, *CuSOD2*), APX (cytosolic enzyme APX, *APX_Cyt_*; chloroplast APX, *APX_Chl_*; stromatic APX, *APX_t+s_*), and CAT presented a common pattern of marked up-regulation at 42/30 °C. Among SOD, APX and CAT genes, *APX_Cyt_* was the most strongly up-regulated one in Geisha 3 (70-fold) and, especially, in Marsellesa (104-fold), whereas in Hybrid plants, the greatest up-regulation was observed for *CuSOD2* (55-fold; Table 2). *CAT* was always significantly up-regulated under heat (between 2-fold in Marsellesa and 6-fold in Geisha 3) but was the less up-regulated of this group of genes encoding for antioxidative enzymes. By Rec14, the transcripts abundance of all studied genes (but VDE) declined in comparison with 42/30 °C, but significantly higher values than the control were still found in all cases, mainly in the antioxidative enzymes (Table 2).

Under heat, eCO_2_ strongly attenuated the observed up-regulation under aCO_2_, for all studied genes and in the three genotypes. That eCO_2_ impact occurred at 37/28 °C (except for *HSP70*, *ELIP*, *APX_Chl_*, *APX_t+s,_* and *VDE* in Geisha 3, and *APX_Chl_* and *VDE* in Hybrid) and 42/30 °C. However, a significant up-regulation was still usually observed in eCO_2_ plants at supra-optimal temperatures when compared with their values at 25/20 °C, meaning that under eCO_2_, gene expression reinforcement still occurred, although to a lesser extent than at aCO_2_. Among the largest gene expression attenuations observed, *CuSOD1, CuSOD2*, and *APX_Cyt_* stood out in the three genotypes, both under 37/28 °C and/or 42/30 °C. Such declines were frequently to values close to or lower than half (Table 2), with striking reductions of 88% (*CuSOD2*, Hybrid, 37/28 °C), 78% (*APX_Cyt_*, Marsellesa, 37/28 °C), and 75% (*APX_Cyt_*, Geisha 3, 42/30 °C) when compared with their respective aCO_2_ counterparts at the same temperature. Additionally, in Rec14, eCO_2_ plants still maintained lower expression values than the aCO_2_ plants but were closer to their controls, suggesting a better recovery to the initial status.

## 3. Materials and Methods

### 3.1. Plant Material, Growth Conditions and Experimental Design

For these experiments were used plants from *Coffea arabica* L. cv. Geisha 3, cv. Marsellesa and their hybrid (Geisha 3 × Marsellesa), which result from breeding efforts to find new cultivars (and in this case also their hybrid) to be used under shaded environments, and to better cope with the new climate conditions estimated to occur along this century. The applied experimental design was similar to that described in [34], with minor modifications. Following a completely randomized design, potted plants (6 to 8 per treatment) were grown in 20 L pots from the seedling stage until ca. two years of age in walk-in growth chambers (EHHF 10000; ARALAB, Albarraque, Portugal) under controlled environmental conditions of temperature (25/20 °C, day/night), irradiance (ca. 700–800 μmol m^−2^ s^−1^), air humidity RH (70%), and photoperiod (12 h), and either ambient (400 μL CO_2_ L^−1^, aCO_2_) or elevated (700 μL CO_2_ L^−1^, eCO_2_). For the entire experiment, the plants were kept under well-watered conditions (predawn water potential higher than −0.3 MPa), adequate mineral nutrient supply (provided as in [59]), and without restrictions as regards space for root growth, as judged by visual examination at the end of the experiment after removing the plants from pots.

### 3.2. Temperature Rise Implementation

Plants were submitted to a gradual temperature increase to allow them to express their potential acclimation capability. The temperature was raised from 25/20 °C (day/night) up to 42/30 °C, at a rate of 0.5 °C day^−1^ (of the diurnal temperature), with 5–7 days of stabilization at 31/25 °C, 37/28 °C, and 42/30 °C to allow for programmed plant material collection. Subsequently, the temperature was readjusted to 25/20 °C and plants were monitored over a recovery period of 4 (Rec4) and 14 (Rec14) days. The control conditions refer to the plants grown at 25/20 °C and aCO_2_.

All measurements were performed on newly matured leaves from the upper third part of the plant canopies, which were flash-frozen in liquid nitrogen and stored at −80 °C until analyses.

The carotenoid concentration and the enzyme activities were given *per* dry weight units (DW) since it is a more stable basis of expression due to the eventual change of leaf hydration status under high-temperature conditions. In this way, the relation between fresh weight (FW) and DW was obtained from the same leaves used for biochemical measurement, similarly to what is established for leaf relative water content (RWC) determinations in *Coffea* spp. [26]. For that, eight foliar discs of 0.5 cm^2^ each were punched from the leaves and FW was immediately determined, whereas DW was obtained after drying the discs at 80 °C for 48 h.

### 3.3. Carotenoids Evaluation

Carotenoids were assessed from three leaf discs (each of 0.5 cm^2^) from 4 to 6 plants per treatment, cut after 1.5–2 h of leaf exposure to diurnal illumination, flash frozen in liquid nitrogen, and stored at −80 °C until analysis. The leaf tissue homogenization and subsequent reversed-phase HPLC analysis were performed as described in [60] using an end-capped C_18_ 5 µm Spherisorb ODS-2 column (250 × 4.6 mm, Waters, Milford, MA, USA). Carotenoid detection was performed at 440 nm in an HPLC system (Waters Alliance e2695, Milford, MA, USA) coupled with a diode-array detector (Waters 2996, Milford, MA, USA). Identification and quantification of each carotenoid were performed using specific standards. The de-epoxidation state, involving xanthophyll cycle components, was calculated as [DEPS = (zeaxanthin + 0.5 antheraxantin)/(violaxanthin + antheraxantin + zeaxanthin)].

### 3.4. Activity of Antioxidative Enzymes

Maximal cellular enzyme activities were assayed using 100 mg fresh weight (FW) of leaf tissue for the most important treatments (the enzyme activity assays were not performed for the 31/25 °C and Rec4 treatments), taken from 3 plants per treatment. All the procedures, from homogenization to enzyme activity measurements, were performed following [26], with some modifications. Briefly, 1 mL of buffer containing 200 mM Tris-HCl (pH 8), 10 mM MgCl_2_ 6H_2_O, 30 mM *β*-mercaptoethanol, 4 mM DTT, 2% Triton X-100, “Complete cocktail EDTA” (2 pills) and 10% glycerol was used, with addition of 1% (1 mL) of polyvinylpolypyrrolidone (PVPP) to each sample in the homogenization phase. The samples were centrifuged (13,000× *g*, 20 min, 4 °C), and the supernatant was used to evaluate enzymatic activities.

Superoxide dismutase (SOD, EC 1.15.1.1) reaction mixture contained 20 mM adrenaline in 50 mM phosphate buffer (pH 7.8) and sodium carbonate buffer (pH 10.4) with EDTA 0.125 mM and 50 μM, and 50 μL of the enzyme extract in a final volume of 1 mL. The activity was spectrophotometrically assessed at 480 nm.

Ascorbate peroxidase (APX, EC 1.11.1.11) reaction mixture contained 20 mM ascorbate and 0.1 mM H_2_O_2_ in 50 mM phosphate buffer (pH 7.8) and 10 μL of the enzyme extract in a total volume of 1 mL. The sample reaction was assessed through the H_2_O_2_-dependent oxidation of ascorbate at 290 nm, using an extinction coefficient of 2.8 mM^−1^ cm^−1^ for calculations.

Glutathione reductase (GR, EC 1.6.4.2) reaction mixture contained 0.15 mM NADPH, 0.5 mM oxidized glutathione (GSSG), 3 mM MgCl_2_ in 50 mM phosphate buffer (pH 7.8), and 10 μL of enzyme extract in a total volume of 1 mL. The activity was evaluated through the NADPH oxidation rate at 340 nm, using an extinction coefficient of 6.22 mM^−1^ cm^−1^ for calculations.

Catalase (CAT, EC 1.11.1.6) reaction mixture contained 40 mM H_2_O_2_ in 50 mM phosphate buffer (pH 7.8) and 10 μL of the enzyme extract in a total volume of 1 mL; activity was evaluated through the rate of H_2_O_2_ consumption at 240 nm, using an extinction coefficient of 3.94 mM^−1^ cm^−1^.

### 3.5. Expression of Genes Associated with Antioxidant and Protective Molecules

Genes encoding antioxidant and other protective proteins were selected based on [26,33] for real-time qPCR studies, using malate dehydrogenase (MDH) and Ubiquitin-conjugating enzyme E2 (*UBQ*), which were found to be among the most stable pair of genes to be used as reference genes for the studied conditions of temperature and [CO_2_] [39]. All primer sequences are presented in Table 3.

Total RNA was isolated from 100 mg of frozen material taken from 3 plants per treatment and processed as described in [35], using the innuPREP Plant RNA Kit (Analytik Jena, Jena, Germany) following the manufacturer’s protocol. The intactness of the extracted RNA was verified by electrophoresis on a 1.5% agarose gel by evaluating the integrity of the 28S and 18S ribosomal RNA bands and the absence of smears. cDNA was synthesized from 1 µg total RNA using the SensiFAST^TM^ cDNA Synthesis kit (Meridian BioScience, Cincinnati, OH, USA), according to the manufacturer’s recommendations. The presence of a single amplification product of the expected gene size was verified by electrophoresis on a 1.5% agarose gel. RT-qPCR reactions were prepared using the SensiFAST^TM^ SYBR No-ROX kit (Meridian BioScience, Cincinnati, OH, USA) according to the manufacturer’s protocol. One negative was included for each primer pair, in which cDNA was replaced by water. Reactions were carried out in 96-well plates using a qTOWER 2.2 Thermal Cycler (Analytik Jena, Jena, Germany) using the following parameters: hot start activation of the Taq DNA polymerase at 95 °C for 10 min, followed by 40 cycles of denaturation at 95 °C for 15 s, annealing at 60 °C for 30 s, elongation at 72 °C for 30 s. A melting curve analysis was performed at the end of the PCR run by a continuous fluorescence measurement from 55 °C to 95 °C with sequential steps of 0.5 °C for 15 s (single peaks were obtained). Three technical replicates were used for each biological replicate.

### 3.6. Statistical Analysis

Data were analyzed using two-way ANOVA (*p* < 0.05) to evaluate the differences between the two atmospheric [CO_2_] (aCO_2_ or eCO_2_) or between the different temperature and recovery treatments (25/20 °C, 31/25 °C, 37/28 °C, 42/30 °C, Rec4, Rec14), and their interaction, followed by Tukey’s HSD test for mean comparisons, except when otherwise stated. The ANOVA for each parameter was performed independently for each of the studied genotypes. For gene expression analysis, the relative expression ratio of each target gene was quantified based on its real-time PCR efficiencies and the crossing point (CP) difference of the unknown sample versus the control (25/20 °C, 400 μL CO_2_ L^−1^ air) within each genotype, as described in [33,39], followed by the same statistical procedure described above. Data were analyzed using Statistica, v8 (StatSoft, Tulsa, OK, USA).

## 4. Discussion

### 4.1. Photoprotective Pigments

#### 4.1.1. Xanthophylls

Under the control temperature, eCO_2_ by itself did not significantly alter the level of most xanthophylls in Geisha 3 and Hybrid plants but promoted greater concentrations of several xanthophylls in Marsellesa (neoxanthin, lutein, the pool of V + A + Z involved in the xanthophyll cycle; Table 1), as also reported in other coffee genotypes [34], revealing its potential photoprotective capability in Marsellesa leaves, even in the absence of stressful conditions. By contrast, under aCO_2,_ the rise of temperature to 37/28 °C promoted an increase in the concentration of several carotenoids (particularly neoxanthin and lutein) in all genotypes, although especially in Geisha 3 and Hybrid plants, as also reflected in a global reinforcement of total carotenoids. Still, both zeaxanthin and DEPS values were not significantly altered in all genotypes, which would be linked to the maintenance of the use of energy through photosynthesis at this temperature [12,34]. This was in line with the moderate rise of zeaxanthin and DEPS under 42/30 °C when the photochemical use of energy (and net photosynthesis) was strongly depressed (data not shown). Such rise in zeaxanthin (and DEPS) were also observed in *C. arabica* cv. Icatu and cv. IPR108 at 42/34 °C, irrespective of [CO_2_], when the photochemical use of energy was compromised [12,34], and plant acclimation (namely of photosystems, electron carriers, and chloroplast membranes) strongly depended on photoprotective thermal dissipation and antioxidative mechanisms [33]. The presence of adequate levels of xanthophylls, and especially zeaxanthin and lutein (but also of *β*-carotene), in the light-harvesting complexes (LHC), are associated with a higher capability of reducing excess excitation energy through thermal dissipation, thus preventing the formation of highly reactive molecules of oxygen and Chl [61,62].

Notably, the eCO_2_ plants showed mostly stable carotenoid values up to 37/28 °C or even 42/30 °C. This suggests a lower need for thermal dissipation when compared with aCO_2_ counterparts, likely associated with the maintenance of a greater photosynthetic functioning due to a greater CO_2_ availability at the chloroplast level and the associated lower energy excess. Furthermore, at 42/30 °C, the Marsellesa plants showed a better potential photoprotective capability due to greater concentrations of lutein and the pool of V + A + Z. Still, this was accompanied by a lowered DEPS value (as also in the other two genotypes) as compared with their aCO_2_ plants. Zeaxanthin was reported to rise (as well as DEPS) in coffee plants exposed to conditions that strongly depress the photochemical use of energy [26,32] since it is associated with a higher need for thermal dissipation and prevention of lipid peroxidation by removing the epoxy groups from the oxidized double bonds of thylakoid fatty acids [63,64]. Therefore, our findings suggest that under the imposed conditions, zeaxanthin was not needed in these plants, although we cannot discard that they were unable to convert violaxanthin into zeaxanthin through VDE action, both of which were suggested by the maintenance or down-regulation of *VDE* (Table 2). Notably, for Hybrid (regardless of [CO_2_]) and Marsellesa (aCO_2_) plants, the need for energy dissipation persisted by Rec4 since zeaxanthin concentration (and DEPS) increased in comparison with both 25/20 °C and 42/30 °C values. Furthermore, Geisha 3 plants showed a higher DEPS value at 42/30 °C, although associated with the maintenance of zeaxanthin and a declined V + A + Z pool, thus supporting the view of an incomplete ability of these genotypes to cope with heat at the xanthophyll cycle level.

Neoxanthin and lutein are found in the periphery of the LHC of photosystems, and they play important functions as they maintain the correct assembly and stability of antenna proteins [65]. In addition, the lutein-epoxide cycle and neoxanthin (as well as *β*-carotene) are quenchers of ^3^Chl* and ^1^O_2_, thus, scavenging these important lipoperoxidation initiators [66,67,68,69,70]. In fact, besides the structural contribution to the antenna complexes, neoxanthin can influence energy harvest, transfer, and dissipation in the photosystem, with implications for photoprotection, namely by increasing the efficiency of ^3^Chl* quenching by lutein, thus contributing to preventing ROS formation and the consequent photoinhibition [71]. Still, neoxanthin did not significantly differ between [CO_2_] treatments and was mostly stable within each [CO_2_] treatment except in Hybrid (increase by 37/28 °C under aCO_2_) and Geisha 3 plants (rise at 31/25 °C and 37/28 °C). By contrast, only the eCO_2_ plants of Marsellesa at 42/30 °C displayed a greater concentration of lutein (than their aCO_2_ counterparts) which likely strengthened the thermal dissipation protection of photosystems against photooxidation in those plants. In fact, the eCO_2_ positive impact on the performance of the photosynthetic apparatus and stress defenses was reported at physiological and molecular levels in studies involving other *C. arabica* and *C. canephora* genotypes, where a significant up-regulation of photosynthetic, antioxidant, and lipid metabolism genes and/or proteins was found under eCO_2_ [28,29,35]. That ultimately mitigated the harsh effects of drought [27,28,29] and heat [34,36], supporting the maintenance of higher photosynthetic performance under eCO_2_ in the studied coffee genotypes.

#### 4.1.2. Carotenes

The *α*- and *β*-carotenes are accessory pigments found in the reaction centers and core antennae of PSI and PSII [72]. Besides the function of absorbing light (especially blue light), carotenes, and especially *β*-carotene, have important photoprotective functions. The latter has the ability to protect lipid components of membranes, and chlorophyll *a* from oxidation, quenching ^1^O_2_ and ^3^Chl* by forming triplet carotenes that dissipate energy through heat [31,73,74]. Furthermore, *β*-carotene protects the cytochrome *b_6_f* complex from photobleaching promoted by ^1^O_2_ [75], and decreased levels of this pigment in the reaction centers have been associated with higher vulnerability to photodamage [74]. The carotenoid biosynthesis pathway has been previously found to be significantly enriched in *C. canephora* cv. CL153 plants grown under eCO_2_ [35], although without a corresponding carotenoid rise [33]. Such lack of impact of the single exposure to eCO_2_ was also found in the present work, except in Marsellesa, which was the only genotype to present a significant rise (55%) in total carotenoids, together with rising tendencies of *α*- and *β*-carotenes (and a 53% rise for (*α* + *β*) carotene). Therefore, Marsellesa plants would have a better potential photoprotection capability (in addition to the already mentioned lutein significant rise), namely against ^3^Chl* and ^1^O_2_ [66,67,68,70].

Regarding the single temperature implementation, *α*- and/or *β*-carotenes tended to have greater concentrations at 31/25 °C and 37/28 °C, similarly to other pigments. That was accompanied by a tendency to higher (*α*/*β*) carotene ratios in all genotypes. However, Marsellesa plants were the only ones to show maximal concentrations of (*α* + *β*) carotenes at 37/28 °C, thus supporting a greater protective function at this moderately high temperature when the other two genotypes showed already a tendency to decline. However, at 42/30 °C, the *α*- and *β*-carotenes strongly declined (as compared to the values at 37/28 °C), especially the first, which was reflected in strong reductions in the (*α*/*β*) carotene ratios of all genotypes, as also found in other genotypes submitted to high temperatures [33]. Although the strong concomitant decline of *β*-carotene suggested that the photosynthetic apparatus was largely affected, the decrease in (*α*/*β*) carotene ratio was interpreted as reflecting a protective mechanism against the energy in excess under cold conditions in *Coffea* sp. [31]. This view agrees with the maintenance of lowered (*α*/*β*) carotene ratio values, even if *β*-carotene and (*α* + *β*) carotene values tended to recover (e.g., in Marsellesa) by Rec14, irrespective of genotype and [CO_2_]. Contrasting with aCO_2_ plants, there were remarkable stability of *α*-, *β*- and (*α* + *β*) carotenes in the eCO_2_ plants from 25/20 °C up to 42/30 °C, thus suggesting the maintenance of photosynthetic structures to which these pigments are associated, e.g., the photosystems. The decline in (*α*/*β*) carotene in Marsellesa at 42/30 °C could suggest a higher thermotolerance capability at the harshest temperature, in line with the finding of a greater potential functioning of the photosynthetic apparatus (evaluated through photosynthetic capacity, A_max_—data not shown). However, in the present case, such (*α*/*β*) carotene decline resulted from a tendency to *α*-reduction and not from an increase in *β*-carotene (which was maintained), contrary to what was found in *C. arabica* cv. Icatu under heat stress, where *β*-carotene showed its maximal values under 42/34 °C as compared to their initial values at 25/20 °C [33,34]. Nevertheless, as compared with their aCO_2_ counterparts, under 42/30 °C, only the eCO_2_ plants of Marsellesa displayed simultaneously greater concentrations of *β*-carotene, lutein, and (V + A + Z), thus denoting a greater potential for photoprotection of the photosynthetic apparatus acting in a complementary way [33].

### 4.2. Antioxidant Enzyme Responses and the Associated Gene Expression

The studied enzymes play important roles in oxidative stress control since they react with several ROS, such as O_2_^●−^ (SOD) and H_2_O_2_ (APX, CAT). The ROS control is greatly accomplished, namely, through the ascorbate-glutathione cycle, with the participation of enzymes such as SOD, APX, and GR, complemented with extra-chloroplast scavenging systems, such as CAT [25,67,76].

Although the single effect of eCO_2_ on the antioxidant enzymes was genotype- and enzyme-dependent, the studied enzyme activities were always maintained or reduced, as compared with the aCO_2_ plant counterparts, with the only exception of APX in Geisha 3, which showed a significant rise. In fact, in the Hybrid plants, no significant changes were observed for all these enzymes, whereas in Marsellesa, SOD and GR activity values declined (Figure 1 and Figure 2). This low responsiveness to eCO_2_ was in accordance with the absence of significant expression changes in *SODs*, *APXs*, and *CAT* (Table 2), showing that under non-stressed conditions, the eCO_2_ had a low, if any, impact on the expression of these genes. This was also in line with the minor changes detected in the primary metabolite profile of *C. canephora* cv. CL153 and *C. arabica* cv. Icatu genotypes are grown under eCO_2_ and control (well-watered and 25 °C) conditions [77]. Our present results also revealed a genotype-dependent response to eCO_2_ among *C. arabica* genotypes but contrasted with previous findings in the above-mentioned genotypes where it was highlighted that eCO_2_ promoted a significant up-regulation of a considerable number of genes related to photosynthetic, antioxidant, and lipid metabolism. These supported the maintenance of increased photosynthetic potential promoted by eCO_2_ and the absence of photosynthesis down-regulation [35]. In fact, single eCO_2_ promoted moderate responsiveness regarding the antioxidative response, which was much lower than the one felt under drought [78] or heat constraints [28,29,36] in those same *C. canephora* and *C. arabica* genotypes.

Contrasting with the single eCO_2_ impact, the single exposure to supra-optimal temperatures was a strong driver of response since it greatly promoted changes in the activity of APX, CAT, and GR (although not of SOD; Figure 1 and Figure 2) and up-regulated genes associated with the antioxidant enzymes (*CAT*, *CuSODs*, *APXs*; Table 2) at 37/28 °C. CAT and APX activities increased in the three genotypes under aCO_2_, reinforcing the potential for H_2_O_2_ control and reflecting a clear response toward the acclimation of the three genotypes. This agrees with the increases in the activity of APX, GR, and CAT under aCO_2_ in *C. canephora* cv. CL153, under eCO_2_ of GR and CAT in *C. arabica* cv. Icatu, as well as of Cu, Zn-SOD and CAT in *C. arabica* cv. IPR108 up to 37 °C [33]. In fact, thermotolerance can be improved by up-regulating gene expression and protein levels of ROS-scavenging enzymes [79,80]. Furthermore, the key protective role of the antioxidant defense system in coffee plants was previously reported to increase under several environmental stress conditions. For instance, Cu, Zn-SOD, and APX activities were enhanced in some coffee genotypes subjected to a gradual cold treatment [24] since an efficient ROS control is crucial to the acclimation response of *C. arabica* and *C. canephora* cultivars exposed to single and combined drought and cold conditions [26]. As stated above, our findings are consistent with the strong up-regulation of genes coding for CAT and APX enzymes at 37/28 °C, which was greater under aCO_2_ than under eCO_2_, both for the enzyme activity and for gene expression. On the other hand, SOD activity tended to decline (significantly in Hybrid plants at 37/28 °C), although the expression of SOD genes was markedly up-regulated. In fact, although they were often coherent, transcript level changes of genes coding for antioxidant enzymes were not always fully in line with the enzyme’s activity pattern. The notion that gene expression does not always perfectly follow biochemical patterns has been previously mentioned in other studies [26,28,33]. For example, at the two highest temperatures, *CuSOD1* and *CuSOD2* were overexpressed in the three genotypes and, to a much greater extent, under aCO_2_ than under eCO_2_. This contrasted with the absence of rises (or even decline) of enzyme activity in most cases, and even with the greater activity at 37/28 °C (Marsellesa and Hybrid) and 42/30 °C (Geisha 3) under eCO_2_ (Figure 1), what can be justified by post-transcriptional and post-translational mechanisms regulating protein synthesis and enzyme functional conformation. This can greatly alter the relation between transcription levels and the biochemical results (e.g., of enzyme activities), thus reinforcing the need for data integration of complementary transcript and other molecular profiling, physiological and biochemical studies to have a clear picture of the real plant response to stress [36].

The difference between enzyme activity and gene expression was particularly striking at 42/30 °C, as also found for chloroplast APX activity and *APX_Chl_* expression in *C. canephora* cv. Conilon [33]. Gene expression associated with antioxidant enzymes showed their maximal absolute values (*CAT*, *CuSODs*, *APXs*) at the highest temperature when the enzyme activities showed quite different and variable impacts. In fact, the activities of GR and CAT were mostly maintained, and the one of APX declined in all genotypes, whereas SOD showed different variations in the genotypes (declined in Hybrid and Marsellesa; maintained in Hybrid), compared to the values at 37/28 °C. Noteworthy is the common response regarding all genotypes and both [CO_2_] (although greater in aCO_2_) at this extreme temperature regarding the activity of CAT and its gene expression. The complementary ability to control the oxidative stress promoted by H_2_O_2_ through CAT reinforcement could be of particular importance due to the concomitant severe decline of cellular APX activity in all genotypes in both [CO_2_] conditions. These findings confirm the high heat sensitivity of APX found in *Coffea* spp. since it was the greatest negatively affected enzyme under 42 °C [33]. Furthermore, it points to a change of H_2_O_2_ control from chloroplast APX [33] or other cellular APXs (Figure 1) until 37/28 °C to an extra-chloroplast control through CAT (which is predominantly located in mitochondria, peroxisomes, and glyoxissomes) at higher temperatures (42 °C). This is of recognized great importance since H_2_O_2_ is capable of diffusing passively across membranes, turning the extra chloroplastic scavenging systems into important H_2_O_2_ detoxification pathways [24,67,81,82].

Regarding GR, the higher (Geisha 3 and Hybrid) or stable (Marsellesa) activity at the two highest temperatures will contribute to regenerating GSH and indirectly ascorbate in the ascorbate-glutathione cycle [67,83,84]. In most cases, a significant up-regulation both at 37/28 °C and 42/30 °C was observed in eCO_2_ plants, as compared with their values at 25/20 °C. However, under such supra-optimal temperatures, the eCO_2_ greatly reduced the transcript abundance of *CAT*, *CuSODs*, and *APXs*, frequently below 50%, with striking attenuations in some cases (e.g., to 12% in *CuSOD2*, Hybrid, 37/28 °C), compared with their aCO_2_ counterparts. Notably, among the studied enzymes, the activity of CAT was the best example of an eCO_2_-induced down-regulation in the three genotypes (Figure 2). Similarly, the activity of APX also declined in Geisha 3 (42/30 °C), Marsellhesa, and Hybrid (both at 37/28 °C). Higher gene expression and enzyme activity under aCO_2_ (than at eCO_2_) suggests a greater presence of ROS and, therefore, a higher need to control them, namely of H_2_O_2_, which is a known signaling molecule that also triggers the expression of genes encoding its scavenging enzymes [82,85]. The relaxation of part of the antioxidant system (considering both the activity of some enzymes and the gene expression of this type of enzymes) under eCO_2_ can be interpreted as reflecting a lower need for a robust antioxidant system [33,86]. That is a consequence of the presence of higher C-assimilation associated with greater photochemical use of energy (data not shown [34]) and minor photorespiration, the latter directly decreasing H_2_O_2_ production [87,88]. This ultimately reduces the energy and H_2_O_2_ pressure on the photosynthetic apparatus, always when compared with aCO_2_ plants. Additionally, Geisha 3 plants under eCO_2_ showed a greater SOD activity than that of aCO_2_ plants, although with a lower gene expression. The overexpression of genes coding for Cu, Zn-SOD, and greater SOD and APX activities has been associated with a greater tolerance to oxidative stress in transgenic tobacco (*Nicotiana tabacum* L.) [89] since a greater SOD activity indicates a stronger control of superoxide radicals, although with an increase in H_2_O_2_ production that could be controlled through APX and CAT action. The antioxidant response under eCO_2_ and its potential mitigating effect under stress conditions have been addressed in other species. The wide variety of obtained results showed not only a strong species-dependency but also varying as well between different cultivars, as reported in this study. For instance, when submitted to heat and drought stress, *Arabidopsis thaliana* plants showed high levels of ascorbate and CAT activity under eCO_2_ [90], whereas in soybean (*Glycine max* (L.) Merr.) declines in SOD, CAT, APX, and GR activities were observed [91], and in alfalfa (*Medicago sativa* L. cv. Aragón) reductions of antioxidant molecules and CAT activity was also reported [86].

After the recovery period (Rec14), with very few exceptions (e.g., CAT in Hybrid under aCO_2_ and in Geisha 3 under eCO_2_), the activity of the antioxidative enzymes declined in comparison with the values observed at 42/30 °C (or were maintained when such values were already similar to controls) regardless of [CO_2_] conditions. That was fully in line with strong declines in transcript abundance of *CAT*, *SODs*, and *APXs*, thus approaching the initial expression levels under control. However, under aCO_2,_ a higher expression of these genes was maintained when compared with eCO_2_. This pattern was particularly evident for *CuSOD2* in Marsellesa and Hybrid and the APX genes in the three genotypes, which points out that a greater need for ROS scavenging is still present in aCO_2_ plants 2 weeks after the end of stress exposure.

### 4.3. Expression of Genes Associated with Other Protective Molecules

Besides the antioxidant enzymes and photoprotective pigments, other protective molecules were assessed through gene expression studies. These molecules are reported to have crucial roles in the maintenance of cellular homeostasis under several environmental stresses, including heat stress [92,93]. Transcript levels of protective molecules, other than the antioxidative enzymes discussed above, i.e., chaperonins 20 and 60 (*Chape20* and *Chape60*), early light-inducible protein (*ELIP*), and 70 kD Heat Shock Proteins (*HSP70s*), showed a similar pattern of variation. These confirmed (1) an absence of response to the single exposure to eCO_2_ in the expression of these genes in the three genotypes (similarly to the findings of [33]), in line with the results of the genes associated with the antioxidative enzymes; (2) a clear overexpression under single heat conditions, with a maximal accumulation of transcripts at 42/30 °C (although similar to the values at 37/28 °C in a few cases) for both aCO_2_ and eCO_2_, but (3) always with greater values under aCO_2_, especially in the Hybrid plants. (4) A subsequent transcriptional decline by Rec14 was observed, although maintaining values above those under the initial control conditions under aCO_2_. Regarding the specific roles of the proteins encoded by these genes, chaperonins ensure the correct folding of new proteins, especially plastid proteins (e.g., RuBisCO), thus playing an important role in heat stress tolerance [94,95,96]. *ELIPs* are found in thylakoid membranes and protect plants under different environmental stresses since they can participate in the antioxidative stress response by dissipating excess energy and preventing the formation of radicals [97,98,99]. As for HSP70s, although in this study, the maximum transcript accumulation was found at 42/30 °C, the greatest abundance of the protein was previously found at 37 °C [33]. These proteins are considered one of the most important protective molecules in plant responses to stress [95,100,101]. Furthermore, previous studies in *Coffea* spp. showed that the HSP70 protein synthesis is among the earlier responses to high temperatures [33], but also to moderate and severe drought, its presence further amplified under the superimposition with eCO_2_ [29]. In the present study, a lower *HSP70* expression under eCO_2_ might also point to a lower need for stress protection and enhanced thermotolerance in these coffee plants since the up-regulation of *HSP70* genes and the greater presence of this protein are usually found under stress conditions [36,95]. Furthermore, HSP70s are involved, among other roles, in PSII repair, and a positive correlation between the expression of the gene encoding for APX and heat shock transcription factors (HSFs) was reported in transgenic plants of *Arabidopsis* under heat stress [102]. As terminal components in the signal transduction chain triggered by heat stress, HSFs bind to the heat shock elements (HSEs) involved in downstream heat-inducible genes, playing a central role in the heat response stress [103]. Thus, a greater presence of HSP70 could also contribute to protecting coffee plants from oxidative stress. Overall, the lower gene expression of these protective molecules (HSP70, ELIP, Chape20; Chape60) under eCO_2_ plants when compared with their aCO_2_ counterparts reinforces the suggestion of a lower need for protecting photosynthetic components from photoinhibition due to increased use of energy through photochemical processes (and lower photorespiration rate), as reflected in the observed photosynthetic performance under eCO_2_ [34,53], which is the better photoprotective mechanism against the build-up of energy overpressure in the chloroplast structures [34].

Our findings highlight that, although, with similar patterns of response, a genotype-dependent response to heat and/or eCO_2_ was clear, both among the *C. arabica* genotypes studied here or regarding previously studied ones where a stronger response relative to protective molecules (e.g., HSP70) was observed [27,36]. Such relevant differences were also found between *C. arabica* and *C. canephora* genotypes [28] and strongly highlight the need to search for thermotolerance biomarkers to be used in breeding programs [54]. Also, some “hybrid vigor” might occur in the studied Hybrid plants when compared to their parental genotypes. This was supported by the greatest responsiveness of all of the studied genes associated with proteins linked to protective roles. Still, that was clear only in aCO_2_ plants at 42/30 °C, which might also suggest a greater need for protective mechanisms, thus needing further studies to understand if this wide up-regulation configures a greater vigor or, by opposition, a stronger sensitivity to the imposed conditions. Both issues regarding the genotype-dependent response to heat and/or eCO_2_ and the “hybrid vigor” potential strongly advise the implementation of accurate breeding and selection programs to get new coffee genotypes. These, together with the use of adequate crop management practices, such as agroforestry [40,104], will be decisive in guaranteeing the environmental and economic sustainability of this crop.

## 5. Conclusions

Overall, eCO_2_ alone barely altered most protective components as compared with aCO_2_. Most xanthophylls were maintained in Geisha 3 and Hybrid plants, but Marsellesa tended to rise the concentration of neoxanthin, lutein, the pool of V + A + Z, (*α* + *β*) carotene, and total carotenoids, improving its photoprotective ability. The enzyme activities did not increase upon the single effect of eCO_2_ (with the exception of APX in Geisha 3), in line with the absence of significant modifications of genes associated with antioxidant enzymes and other protective proteins, always relative to aCO_2_ plant counterparts.

Temperature was the main driver of plant response at 37/28 °C and 42/30 °C., At 37/28 °C several carotenoids (particularly neoxanthin and lutein) increased in all genotypes, whereas at 42/30 °C a moderate increase of zeaxanthin and DEPS was found, likely contributing to preventing the formation of highly reactive molecules of O_2_ and Chl when the photochemical use of energy would be quite depressed. Marsellesa showed greater additional carotenoids photoprotection at 37/28 °C than the other genotypes, including maximal (*α* + *β*) carotene concentration, although the latter declined in all genotypes at 42/30 °C.

With the exception of *VDE* (down-regulated with heat), all genes encoding for antioxidative enzymes (*CAT*, *CuSODs*, *APX*) or other protective proteins (*HSP70*, *ELIP*, *Chape20*, *Chape60*), exhibited a high responsiveness to single heat in all genotypes, with a strong up-regulation at 37/28 °C. This was usually even greater at 42/30 °C, with the genes associated with antioxidative enzymes showing the greatest transcripts abundance. Accordingly, 37/28 °C greatly promoted the activity of APX and CAT in all genotypes (likely controlling H_2_O_2_ presence), and of GR (except in Marsellesa), but not of SOD (that tended to decline despite the large increase of SODs transcripts). CAT activity deserves a special mention since it increased even at 42/30 °C, thus compensating for the strong APX activity decline at this temperature in all genotypes. An increased potential thermotolerance could arise also through the reinforcement of HSP70, ELIP, Chape20, and Chape60 molecules, as supported by a large up-regulation of their associated genes. Hybrid plants showed the greatest gene up-regulation of most genes under 42/30 °C.

In contrast with the low responsiveness to single eCO_2_ exposure, the superimposition with high temperatures revealed an interaction between heat and eCO_2_. The eCO_2_ plants showed mostly stable carotenoid values up to 37/28 °C or even 42/30 °C, but only Marsellesa plants denoted greater photoprotective capabilities at 42/30 °C, through greater concentrations of lutein, V + A + Z, and *β*-carotene, as compared their aCO_2_ plants. The eCO_2_ attenuated the gene up-regulation observed in aCO_2_ plants under heat and lower lowered the antioxidative system (CAT activity is the best example), pointing to a lesser need for ROS control (e.g., H_2_O_2_) supported by the persistence of photochemical use of energy and low photorespiration in eCO_2_ plants.

An incomplete recovery by Rec14 was suggested in the Hybrid and Marsellesa (regardless of [CO_2_]) due to greater zeaxanthin (and DEPS) values than in control, reflecting the persistence of a thermal dissipation need. Although most antioxidant enzyme activities declined towards their initial values, the partial recovery in all genotypes was further pointed out by the maintenance of up-regulation of all genes, usually much greater under aCO_2_ than in eCO_2_ counterparts, what denoted a better/faster recovery under eCO_2_. Still, higher levels of antioxidant components (molecules and gene expression) in aCO_2_ plants can also be seen as a protection strategy, allowing the plants to better endure new stress events, whereas, in eCO_2_ counterparts, such a role could be performed by the greater photochemical use of energy.

Altogether, *C. arabica* plants responded to heat and/or eCO_2_ in a genotype-dependent manner, with a greater acclimation potential in Marsellesa. Heat was the main response driver for the addressed protective molecules and genes, whereas eCO_2_ alone did not greatly altered plant status but usually attenuated the heat response, likely supported by a greater use of energy through photochemistry. The importance of the acclimation process and their responsiveness turn these molecules/genes useful biomarkers to breeding and selection programs, which should also explore the heterosis advantages that might arise from select hybrid crosses. Together with adequate management practices, these can help this crop to thrive under global warming conditions.

## Figures and Tables

**Figure 1 plants-11-02702-f001:**
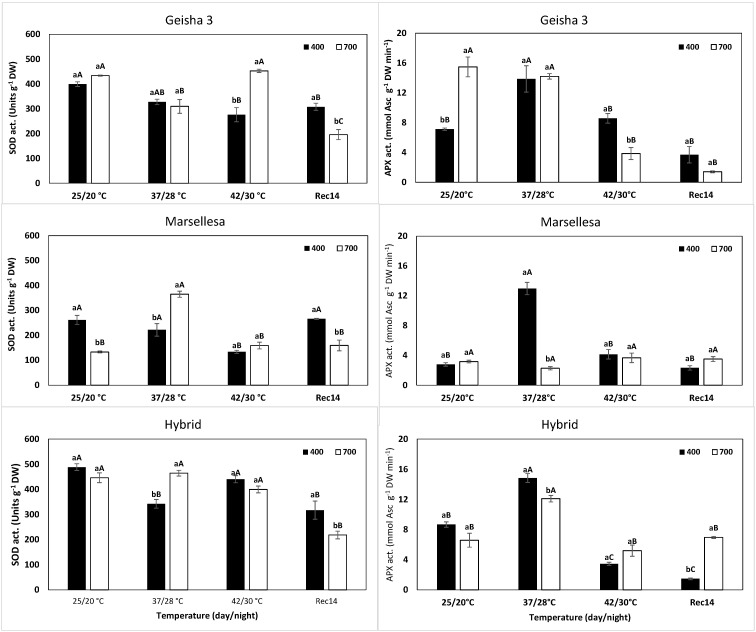
Changes in leaf cellular activities of superoxide dismutase (SOD) and ascorbate peroxidase (APX) enzymes in *C. arabica* cv. Geisha 3, cv. Marsellesa and their Hybrid (Geisha 3 × Marsellesa) plants grown under 400 or 700 μLCO_2_ L^−1^ at control (25/20 °C, day/night), submitted to supra-optimal temperatures (37/28 °C, 42/30 °C), and after 14 days of recovery (Rec14). For each enzyme, the mean values ± SE (*n* = 3 plants) followed by different letters express significant differences between CO_2_ treatments for each temperature, separately for each genotype (a, b), or between temperatures for the same CO_2_ treatment (A, B, C), always separately for each genotype, where a > b and A > B > C.

**Figure 2 plants-11-02702-f002:**
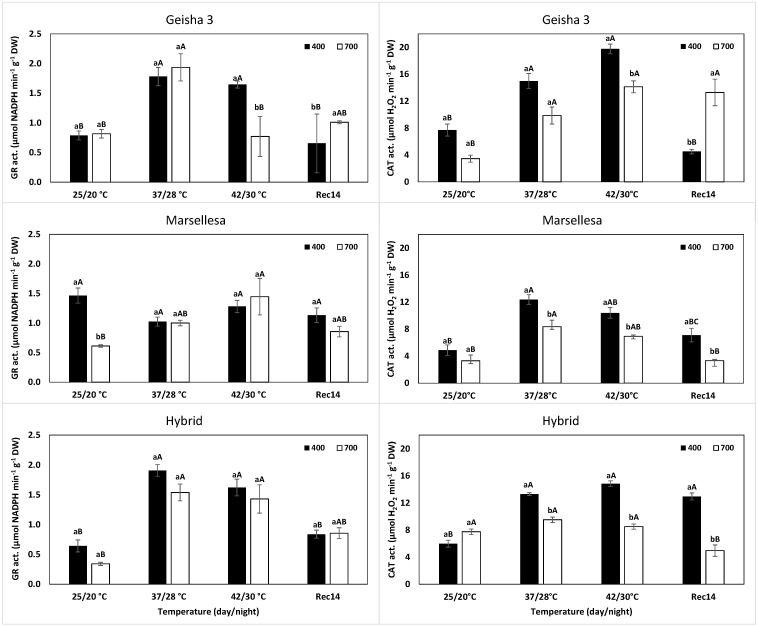
Changes in leaf cellular activities of glutathione reductase (GR) and catalase (CAT) enzymes in *C. arabica* cv. Geisha 3, cv. Marsellesa and their Hybrid (Geisha 3 × Marsellesa) plants grown under 400 or 700 μLCO_2_ L^−1^ at control (25/20 °C, day/night), submitted to supra-optimal temperatures (37/28 °C, 42/30 °C) and after 14 days of recovery (Rec14). For each enzyme, the mean values ± SE (*n* = 3 plants) followed by different letters express significant differences between CO_2_ treatments for each temperature, separately for each genotype (a, b), or between temperatures for the same CO_2_ treatment (A, B, C), always separately for each genotype, where a > b and A > B > C.

**Table 1 plants-11-02702-t001:** Leaf carotenoids concentration (mg g^−1^ dry weight, DW) in *C. arabica* cv. Geisha 3, cv. Marsellesa and their Hybrid (Geisha 3 × Marsellesa) plants grown under 400 or 700 μLCO_2_ L^−1^ at the control temperature (25/20 °C, day/night), submitted to supra-optimal temperatures (31/25 °C, 37/28 °C, 42/30 °C), and after 4 (Rec4) and 14 (Rec14) days of recovery. For each parameter, the mean values ± SE (*n* = 4–6 plants) followed by different letters express significant differences between CO_2_ treatments for each temperature (a, b) or between temperatures for the same CO_2_ treatment (A, B, C, D), always separately for each genotype, where a > b and A > B > C > D.

Pigment	Genotype	[CO_2_] (µL L^−1^)	Temperature (Day/Night)					
			25/20 °C	31/25 °C	37/28 °C	42/30 °C	Rec4	Rec14
**Neoxanthin** **(mg g^−1^ DW)**	**Geisha 3**	400	0.195 ± 0.004 aC	0.293 ± 0.014 aA	0.269 ± 0.004 aAB	0.178 ± 0.012 aC	0.222 ± 0.010 aBC	0.201 ± 0.028 aC
		700	0.180 ± 0.019 aA	0.168 ± 0.020 bA	0.227 ± 0.025 aA	0.197 ± 0.006 aA	0.170 ± 0.014 aA	0.182 ± 0.020 aA
	**Marsellesa**	400	0.270 ± 0.039 bA	0.298 ± 0.041 aA	0.315 ± 0.015 aA	0.215 ± 0.042 aA	0.329 ± 0.019 aA	0.234 ± 0.022 aA
		700	0.404 ± 0.024 aA	0.363 ± 0.033 aA	0.368 ± 0.034 aA	0.321 ± 0.014 aA	0.304 ± 0.012 aA	0.295 ± 0.014 aA
	**Hybrid**	400	0.245 ± 0.003 aB	0.291 ± 0.025 aAB	0.346 ± 0.030 aA	0.219 ± 0.022 aB	0.219 ± 0.015 aB	0.236 ± 0.016 aB
		700	0.251 ± 0.027 aAB	0.254 ± 0.019 aAB	0.236 ± 0.017 bAB	0.291 ± 0.013 aA	0.178 ± 0.011 aB	0.221 ± 0.011 aAB
**Violaxanthin** **(mg g^−1^ DW)**	**Geisha 3**	400	0.203 ± 0.008 aBC	0.360 ± 0.013 aA	0.247 ± 0.014 aB	0.093 ± 0.010 aD	0.154 ± 0.011 aCD	0.187 ± 0.023 aBC
		700	0.178 ± 0.028 aAB	0.205 ± 0.032 bA	0.221 ± 0.038 aA	0.139 ± 0.004 aB	0.157 ± 0.009 aAB	0.148 ± 0.009 aB
	**Marsellesa**	400	0.299 ± 0.050 aA	0.326 ± 0.050 bA	0.327 ± 0.016 aA	0.132 ± 0.031 bB	0.195 ± 0.008 aAB	0.182 ± 0.026 aAB
		700	0.435 ± 0.032 aA	0.467 ± 0.031 aA	0.385 ± 0.032 aAB	0.276 ± 0.031 aBC	0.219 ± 0.013 aC	0.192 ± 0.027 aC
	**Hybrid**	400	0.270 ± 0.015 aB	0.351 ± 0.044 aAB	0.356 ± 0.019 aA	0.153 ± 0.010 bC	0.154 ± 0.029 aC	0.205 ± 0.019 aBC
		700	0.246 ± 0.027 aAB	0.334 ± 0.014 aA	0.263 ± 0.018 aAB	0.267 ± 0.022 aAB	0.120 ± 0.028 aB	0.169 ± 0.011 aAB
**Antheraxanthin** **(mg g^−1^ DW)**	**Geisha 3**	400	0.058 ± 0.005 aA	0.043 ± 0.010 aA	0.062 ± 0.005 aA	0.048 ± 0.004 aA	0.045 ± 0.002 aA	0.021 ± 0.000 aB
		700	0.059 ± 0.004 aA	0.059 ± 0.003 aA	0.066 ± 0.004 aA	0.040 ± 0.008 aAB	0.026 ± 0.009 aB	0.028 ± 0.003 aB
	**Marsellesa**	400	0.046 ± 0.012 aA	0.044 ± 0.014 aA	0.064 ± 0.007 aA	0.064 ± 0.011 aA	0.073 ± 0.006 aA	0.041 ± 0.009 aA
		700	0.067 ± 0.008 aA	0.043 ± 0.010 aA	0.069 ± 0.016 aA	0.060 ± 0.007 aA	0.048 ± 0.009 aA	0.040 ± 0.008 aA
	**Hybrid**	400	0.039 ± 0.002 aB	0.054 ± 0.015 aAB	0.056 ± 0.012 aAB	0.077 ± 0.013 aA	0.069 ± 0.003 aAB	0.062 ± 0.005 aAB
		700	0.047 ± 0.007 aA	0.038 ± 0.006 aA	0.059 ± 0.008 aA	0.040 ± 0.006 aA	0.042 ± 0.003 aA	0.046 ± 0.004 aA
**Zeaxanthin** **(mg g^−1^ DW)**	**Geisha 3**	400	0.094 ± 0.005 aA	0.030 ± 0.014 aB	0.092 ± 0.014 aA	0.097 ± 0.015 aA	0.049 ± 0.019 aAB	0.008 ± 0.001 aB
		700	0.057 ± 0.002 aAB	0.066 ± 0.006 aAB	0.094 ± 0.010 aAB	0.056 ± 0.017 aAB	0.034 ± 0.006 aB	0.023 ± 0.004 aB
	**Marsellesa**	400	0.015 ± 0.002 aB	0.013 ± 0.003 aB	0.043 ± 0.011 aB	0.097 ± 0.026 aAB	0.144 ± 0.035 aA	0.123 ± 0.058 aA
		700	0.089 ± 0.022 aAB	0.026 ± 0.007 aB	0.018 ± 0.005 aB	0.097 ± 0.033 aAB	0.084 ± 0.036 aAB	0.182 ± 0.073 aA
	**Hybrid**	400	0.051 ± 0.007 aB	0.061 ± 0.011 aB	0.063 ± 0.016 aB	0.114 ± 0.034 aB	0.228 ± 0.040 aA	0.114 ± 0.037 aB
		700	0.066 ± 0.028 aB	0.022 ± 0.008 aB	0.037 ± 0.010 aB	0.029 ± 0.005 aB	0.179 ± 0.054 aA	0.106 ± 0.026 aAB
**V + A + Z** **(mg g^−1^ DW)**	**Geisha 3**	400	0.355 ± 0.013 aAB	0.433 ± 0.031 aA	0.401 ± 0.030 aA	0.239 ± 0.025 aB	0.248 ± 0.020 aB	0.216 ± 0.022 aB
		700	0.380 ± 0.039 aAB	0.343 ± 0.031 aAB	0.381 ± 0.033 aA	0.235 ± 0.028 aB	0.217 ± 0.007 aB	0.198 ± 0.014 aB
	**Marsellesa**	400	0.360 ± 0.061 bA	0.383 ± 0.061 aA	0.433 ± 0.031 aA	0.272 ± 0.044 aA	0.412 ± 0.042 aA	0.346 ± 0.043 aA
		700	0.592 ± 0.051 aA	0.536 ± 0.035 aAB	0.473 ± 0.049 aAB	0.434 ± 0.019 aAB	0.351 ± 0.033 aB	0.414 ± 0.059 aAB
	**Hybrid**	400	0.360 ± 0.020 aA	0.466 ± 0.062 aA	0.475 ± 0.031 aA	0.344 ± 0.050 aA	0.451 ± 0.025 aA	0.381 ± 0.023 aA
		700	0.387 ± 0.043 aA	0.394 ± 0.025 aA	0.360 ± 0.026 aA	0.336 ± 0.025 aA	0.341 ± 0.026 aA	0.321 ± 0.020 aA
**DEPS**	**Geisha 3**	400	0.345 ± 0.014 aAB	0.105 ± 0.030 aC	0.299 ± 0.024 aAB	0.499 ± 0.034 aA	0.272 ± 0.055 aB	0.093 ± 0.013 aC
		700	0.305 ± 0.030 aA	0.282 ± 0.026 aA	0.354 ± 0.049 aA	0.293 ± 0.053 aA	0.212 ± 0.032 aA	0.178 ± 0.012 aA
	**Marsellesa**	400	0.116 ± 0.008 aB	0.098 ± 0.015 aB	0.163 ± 0.021 aB	0.481 ± 0.072 aA	0.402 ± 0.051 aAB	0.320 ± 0.115 aAB
		700	0.202 ± 0.024 aAB	0.088 ± 0.018 aB	0.105 ± 0.017 aB	0.280 ± 0.072 aA	0.257 ± 0.076 aA	0.383 ± 0.124 aAB
	**Hybrid**	400	0.193 ± 0.008 aB	0.189 ± 0.022 aB	0.187 ± 0.020 aB	0.383 ± 0.064 aAB	0.572 ± 0.075 aA	0.352 ± 0.072 aAB
		700	0.203 ± 0.045 aB	0.097 ± 0.021 aB	0.178 ± 0.032 aB	0.145 ± 0.022 bB	0.524 ± 0.120 aA	0.367 ± 0.066 aAB
**Lutein** **(mg g^−1^ DW)**	**Geisha 3**	400	0.601 ± 0.007 aC	0.833 ± 0.027 aB	0.780 ± 0.033 aB	0.756 ± 0.043 aB	1.036 ± 0.058 aA	0.809 ± 0.104 aB
		700	0.534 ± 0.042 aBC	0.503 ± 0.042 bC	0.650 ± 0.052 aAB	0.710 ± 0.032 aA	0.649 ± 0.043 bAB	0.646 ± 0.071 bAB
	**Marsellesa**	400	0.779 ± 0.109 bBC	0.842 ± 0.107 aBC	0.945 ± 0.051 aB	0.612 ± 0.067 bC	1.427 ± 0.045 aA	0.886 ± 0.076 aBC
		700	1.198 ± 0.085 aA	1.010 ± 0.072 aA	1.030 ± 0.081 aA	1.153 ± 0.054 aA	1.148 ± 0.075 aA	1.101 ± 0.075 aA
	**Hybrid**	400	0.731 ± 0.020 aB	0.913 ± 0.069 aAB	1.021 ± 0.068 aA	0.919 ± 0.074 aAB	1.054 ± 0.068 aA	1.026 ± 0.035 aA
		700	0.742 ± 0.105 aB	0.706 ± 0.057 aB	0.683 ± 0.042 bB	1.042 ± 0.049 aA	0.807 ± 0.035 bAB	0.907 ± 0.070 aA
***α*-carotene** **(mg g** **^−1^ DW)**	**Geisha 3**	400	0.129 ± 0.011 aB	0.265 ± 0.014 aA	0.256 ± 0.009 aA	0.074 ± 0.010 aB	0.071 ± 0.010 aB	0.046 ± 0.005 aB
		700	0.120 ± 0.019 aB	0.113 ± 0.030 bB	0.224 ± 0.039 aA	0.094 ± 0.002 aB	0.056 ± 0.012 aB	0.062 ± 0.016 aB
	**Marsellesa**	400	0.225 ± 0.048 aAB	0.279 ± 0.047 aA	0.319 ± 0.034 aA	0.121 ± 0.040 aB	0.187 ± 0.028 aAB	0.109 ± 0.026 aB
		700	0.364 ± 0.036 aA	0.306 ± 0.041 aAB	0.306 ± 0.058 aAB	0.240 ± 0.027 aAB	0.190 ± 0.017 aB	0.117 ± 0.017 aB
	**Hybrid**	400	0.156 ± 0.017 aBC	0.278 ± 0.037 aA	0.255 ± 0.014 aAB	0.107 ± 0.020 bC	0.055 ± 0.010 aC	0.072 ± 0.015 aC
		700	0.205 ± 0.038 aA	0.223 ± 0.041 aA	0.209 ± 0.025 aAB	0.223 ± 0.026 aA	0.060 ± 0.012 aB	0.080 ± 0.009 aB
***β*-carotene** **(mg g** **^−1^ DW)**	**Geisha 3**	400	0.314 ± 0.016 aC	0.513 ± 0.023 aA	0.407 ± 0.009 aB	0.322 ± 0.025 aBC	0.277 ± 0.016 aC	0.291 ± 0.051 aC
		700	0.286 ± 0.016 aA	0.298 ± 0.033 bA	0.374 ± 0.032 aA	0.317 ± 0.018 aA	0.251 ± 0.025 aA	0.254 ± 0.049 aA
	**Marsellesa**	400	0.353 ± 0.070 aAB	0.378 ± 0.075 aAB	0.488 ± 0.027 aA	0.233 ± 0.048 bB	0.439 ± 0.030 aAB	0.367 ± 0.035 aAB
		700	0.522 ± 0.042 aA	0.561 ± 0.046 aA	0.528 ± 0.079 aA	0.483 ± 0.030 aA	0.456 ± 0.015 aA	0.427 ± 0.028 aA
	**Hybrid**	400	0.371 ± 0.013 aAB	0.475 ± 0.047 aA	0.421 ± 0.016 aAB	0.381 ± 0.027 aAB	0.306 ± 0.018 aB	0.361 ± 0.014 aAB
		700	0.289 ± 0.043 aB	0.392 ± 0.016 aAB	0.433 ± 0.034 aA	0.331 ± 0.042 aAB	0.312 ± 0.024 aAB	0.360 ± 0.012 aAB
**(*α* + *β*) carotene** **(mg g** **^−1^ DW)**	**Geisha 3**	400	0.443 ± 0.015 aB	0.778 ± 0.035 aA	0.662 ± 0.008 aA	0.396 ± 0.032 aB	0.348 ± 0.021 aB	0.336 ± 0.056 aB
		700	0.406 ± 0.033 aB	0.411 ± 0.062 bB	0.598 ± 0.071 aA	0.411 ± 0.018 aAB	0.306 ± 0.037 aB	0.317 ± 0.063 aB
	**Marsellesa**	400	0.578 ± 0.115 aAB	0.657 ± 0.121 aAB	0.807 ± 0.060 aA	0.355 ± 0.087 bB	0.626 ± 0.057 aAB	0.476 ± 0.059 aB
		700	0.886 ± 0.057 aA	0.867 ± 0.081 aA	0.834 ± 0.132 aA	0.722 ± 0.033 aA	0.645 ± 0.028 aA	0.544 ± 0.043 aA
	**Hybrid**	400	0.528 ± 0.024 aB	0.753 ± 0.083 aA	0.676 ± 0.022 aAB	0.488 ± 0.046 aBC	0.362 ± 0.028 aC	0.433 ± 0.022 aC
		700	0.494 ± 0.081 aAB	0.615 ± 0.051 aAB	0.642 ± 0.058 aA	0.554 ± 0.045 aAB	0.372 ± 0.019 aB	0.440 ± 0.010 aB
**(*α*/*β*) carotene** **(g g^−1^ DW)**	**Geisha 3**	400	0.426 ± 0.052 aB	0.518 ± 0.018 aAB	0.633 ± 0.035 aA	0.228 ± 0.025 aC	0.259 ± 0.037 aC	0.175 ± 0.024 aC
		700	0.404 ± 0.052 aAB	0.346 ± 0.054 bB	0.576 ± 0.056 aA	0.301 ± 0.022 aBC	0.207 ± 0.034 aC	0.255 ± 0.031 aBC
	**Marsellesa**	400	0.658 ± 0.079 aAB	0.764 ± 0.058 aA	0.642 ± 0.040 aAB	0.424 ± 0.074 aBC	0.411 ± 0.038 aBC	0.271 ± 0.049 aC
		700	0.718 ± 0.077 aA	0.545 ± 0.070 aAB	0.597 ± 0.069 aAB	0.522 ± 0.086 aB	0.415 ± 0.035 aBC	0.267 ± 0.026 aC
	**Hybrid**	400	0.422 ± 0.044 aAB	0.570 ± 0.027 aA	0.611 ± 0.034 aA	0.266 ± 0.033 bB	0.172 ± 0.025 aB	0.199 ± 0.041 aB
		700	0.696 ± 0.033 aAB	0.561 ± 0.086 aAB	0.475 ± 0.034 aB	0.770 ± 0.135 aA	0.210 ± 0.048 aC	0.230 ± 0.030 aC
**Total carotenoids** **(mg g** **^−1^ DW)**	**Geisha 3**	400	1.594 ± 0.032 aC	2.338 ± 0.099 aA	2.113 ± 0.009 aAB	1.569 ± 0.106 aB	1.854 ± 0.103 aB	1.562 ± 0.203 aB
		700	1.500 ± 0.131 aA	1.425 ± 0.154 bA	1.855 ± 0.180 aA	1.553 ± 0.070 aA	1.343 ± 0.088 bA	1.343 ± 0.159 aB
	**Marsellesa**	400	1.987 ± 0.317 bAB	2.180 ± 0.320 aAB	2.500 ± 0.151 aAB	1.688 ± 0.332 aB	2.795 ± 0.138 aA	1.941 ± 0.165 aAB
		700	3.079 ± 0.196 aA	2.777 ± 0.215 aA	2.705 ± 0.290 aA	2.630 ± 0.114 aA	2.448 ± 0.139 aA	2.354 ± 0.168 aA
	**Hybrid**	400	1.864 ± 0.042 aB	2.423 ± 0.227 aAB	2.519 ± 0.123 aA	1.970 ± 0.190 aAB	2.085 ± 0.101 aAB	2.076 ± 0.059 aAB
		700	1.875 ± 0.238 aA	1.969 ± 0.133 aA	1.921 ± 0.138 bA	2.223 ± 0.121 aA	1.698 ± 0.049 aA	1.890 ± 0.088 aA

**Table 2 plants-11-02702-t002:** Real-time PCR expression studies relative to the expression value observed under control conditions of temperature and CO_2_ (25/20 °C, 400 μL CO_2_ L^−1^) from leaves of *C. arabica* cv. Geisha3, cv. Marsellesa and their Hybrid (Geisha 3 × Marsellesa) plants grown under 400 or 700 μL CO_2_ L^−1^ at control (25/20 °C, day/night), submitted to supra-optimal temperatures (37/28 °C, 42/30 °C), and after 14 days of recovery (Rec14). For each gene transcript, the mean values (*n* = 3 plants) followed by different letters express significant differences between [CO_2_] levels for each temperature treatment (a, b) or between temperature treatments for the same CO_2_ treatment (A, B, C, D), always separately for each genotype, where a > b and A > B > C > D.

Genotype	Temperature (Day/Night)	[CO_2_] (µL L^−1^)	*HSP70*		*ELIP*		*Chape20*		*Chape60*		*CAT*		*CuSOD1*		*CuSOD2*		*APX_Cyt_*		*APX_Chl_*		*APX_t_* _+*s*_		*VDE2*	
**Geisha 3**	**25/20 °C**	400	1.00	aD	1.00	aD	1.00	aD	1.00	aD	1.00	aD	1.00	aD	1.00	aC	1.00	aD	1.00	aC	1.00	aD	1.00	aA
		700	0.98	aD	0.96	aC	1.02	aC	1.21	aC	0.98	aB	0.98	aC	0.95	aC	0.88	aD	0.98	aC	1.02	aC	0.99	aA
	**37/28 °C**	400	1.78	aC	2.23	aB	8.22	aB	6.65	aB	3.42	aC	4.55	aB	25.23	aA	48.23	aB	12.31	aB	15.34	aB	0.56	aB
		700	1.76	aB	2.21	aA	4.55	bA	5.67	bB	2.22	bA	3.23	bA	12.24	bA	14.55	bB	10.25	aA	14.24	aA	0.54	aC
	**42/30 °C**	400	4.22	aA	3.45	aA	11.21	aA	9.21	aA	6.56	aA	6.66	aA	28.91	aA	69.89	aA	24.55	aA	26.77	aA	0.98	aA
		700	2.25	bA	2.44	bA	4.55	bA	6.23	bA	2.27	bA	3.56	bA	11.55	bA	17.67	bA	12.34	bA	15.61	bA	0.72	bB
	**Rec14**	400	2.21	aB	1.25	aC	7.72	aC	2.23	aC	4.46	aB	2.23	aC	2.66	aB	26.55	aC	14.55	aB	8.6	aC	0.98	aA
		700	1.24	bC	1.22	aB	2.33	bB	1.27	bC	2.26	bA	1.67	bB	2.21	aB	12.21	bC	8.90	bB	2.24	bB	0.88	aB
**Marsellesa**	**25/20 °C**	400	1.00	aD	1.00	aC	1.00	aC	1.00	aD	1.00	aB	1.00	aC	1.00	aB	1.00	aD	1.00	aD	1.00	aD	1.00	aA
		700	1.02	aC	0.98	aC	0.98	aB	1.02	aC	1.04	aC	0.99	aC	0.92	aB	0.99	aD	1.03	aD	1.05	aD	0.96	aA
	**37/28 °C**	400	2.24	aB	3.33	aA	3.46	aB	3.34	aB	2.61	aA	4.23	aB	14.55	aA	48.91	aB	18.21	aB	22.34	aB	0.97	aA
		700	1.88	bB	2.24	bA	2.23	bA	1.67	bB	2.18	bA	2.29	bB	0.98	bB	10.98	bC	8.23	bB	11.36	bB	0.76	bB
	**42/30 °C**	400	4.33	aA	3.37	aA	6.78	aA	4.54	aA	2.67	aA	8.99	aA	14.67	aA	104.22	aA	24.56	aA	39.31	aA	0.55	aB
		700	2.21	bA	2.27	bA	2.21	bA	2.21	bA	2.17	bA	4.55	bA	1.02	bB	55.66	bA	19.18	bA	22.8	bA	0.43	bC
	**Rec14**	400	1.25	bC	2.21	aB	3.23	aB	1.99	aC	2.41	aA	4.65	aB	14.22	aA	26.71	aC	5.44	aC	6.33	aC	0.98	aA
		700	2.22	aA	1.98	bB	1.12	bB	1.03	bC	1.98	bB	2.33	bB	2.21	bA	14.33	bB	2.23	bC	3.35	bC	0.78	bB
**Hybrid**	**25/20 °C**	400	1.00	aD	1.00	aD	1.00	aD	1.00	aD	1.00	aD	1.00	aD	1.00	aD	1.00	aD	1.00	aD	1.00	aD	1.00	aA
		700	0.97	aC	0.96	bC	0.98	aD	0.98	aD	0.99	aC	0.99	aC	1.13	aC	1.22	aC	1.02	aC	0.98	aD	0.97	bA
	**37/28 °C**	400	6.22	aB	5.66	aB	11.25	aB	8.91	aB	3.44	aB	8.88	aB	35.61	aB	23.56	aB	26.77	aB	32.44	aB	0.94	aA
		700	2.43	bA	2.23	bA	6.78	bB	4.56	bB	2.23	bA	1.44	bBC	4.57	bB	9.21	bB	22.11	aA	14.55	bB	0.96	aA
	**42/30 °C**	400	13.98	aA	8.66	aA	26.79	aA	11.23	aA	5.21	aA	12.34	aA	55.34	aA	36.21	aA	33.72	aA	43.77	aA	0.56	bB
		700	2.49	bA	2.21	bA	9.87	bA	8.86	bA	2.21	bA	5.57	bA	8.87	bA	11.23	bA	21.40	bA	24.78	bA	0.67	aC
	**Rec14**	400	3.44	aC	2.82	aC	5.44	aC	2.45	aC	2.49	aC	4.56	aC	13.44	aC	17.22	aC	14.23	aC	24.21	aC	0.98	aA
		700	1.22	bB	1.45	bB	3.56	bC	1.23	bC	1.89	bB	2.26	bB	1.22	bC	8.54	bB	5.51	bB	11.20	bC	0.77	bB

**Table 3 plants-11-02702-t003:** Selected genes used for real-time qPCR studies related to protective mechanisms and/or oxidative stress control, primer sequences and amplicon size (bp).

Gene Symbol	Gene Description	Primer Sequence (5′–3′)	Amplicon Size (bp)
*HSP70*	Stromal 70 kDa heat shock-related protein, chloroplastic	F: GGGAAGCAATTGACACCAAG	150
		R: AGCCACCAGATACTGCATCC	
*ELIP*	Chloroplast early light-induced protein	F: GCCATGATAGGGTTTGTTGC	101
		R: GTCCCAATGAACCATTGCAG	
*Chape20*	Chloroplast 20 kDa chaperonin	F: GTTAAAGCTGCCGCTGTTG	150
		R: CTCACCTCCTTGAGGTTTCG	
*Chape60*	Mitochondria chaperonin CPN60	F: GGATAGTGAAGCCCTTGC	80
		R: CCCAGGAGCTTTTATTGCAC	
*CAT*	Catalase isozyme 1	F: CTACTTCCCCTCGCGGTAT	150
		R: CTGTCTGGTGCAAATGAACG	
*CuSOD1*	Superoxide dismutase [Cu-Zn]	F: CCCTTGGAGACACAACGAAT	141
		R: GGCAGTACCATCTTGACCA	
*CuSOD2*	Superoxide dismutase [Cu-Zn]	F: GGGGCTCTATCCAATTCCTC	150
		R: GGTTAAAATGAGGCCCAGTG	
*APX_Cyt_*	Cytosol ascorbate peroxidase	F: TCTGGATTTGAGGGACCTTG	108
		R: GTCAGATGGAAGCCGGATAA	
*APX_Chl_*	Chloroplast ascorbate peroxidase	F: CACCTGCTGCTCATTTACG	100
		R: GACCTTCCCAATGTGTGTG	
*APX_t+s_*	Stromatic ascorbate peroxidase (sAPX) mRNA	F: AGGGCAGAATATGAAGGATTGG	112
		R: CCAAGCAAGGATGTCAAAATAGCC	
*VDE2*	Violaxanthin de-epoxidase	F: GGGTTCAAAATGCACAAGACTG	86
		R: CCCTCTTTTACCTCAGGCATTG	

## Data Availability

Data is contained within the article.

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
