# Peer review of "Protective Responses at the Biochemical and Molecular Level Differ between a Coffea arabica L. Hybrid and Its Parental Genotypes to Supra-Optimal Temperatures and Elevated Air [CO2]"

_plants, 2022, doi:10.3390/plants11202702_

Round 1

Reviewer 1 Report

My comments are in the attached file.

Author Response

We thank the reviewer for their comments and suggestions.

We introduced the suggested modifications, but when needed we provide a more complete answer regarding some specific questions.

Title

[CO2] or eCO2?

Answer: since we did not defined what means eCO2 (elevated air CO2 concentration) and because the use of brackets is a wide used convention in chemistry/biology for “concentration”, thus [CO2] is easily recognized as “CO2 concentration”. Therefore, we think that the original way can be maintained.

Material and Methods

Why you used these cultivars?

Answer:

These cultivars are used because they have result from breeding efforts to find new cultivars (and in this case also their hybrid) that can face the new climate conditions. These cultivars are already in use in several coffee producing countries (namely in Asia, Vietnam, and South America, e.g., Nicaragua)

That was introduced introduced at the beginning of 2.1 point.

Point 2.4

“FW”

Answer: “FW” is a widely used convention for “Fresh weight”. We define it the first time we use it

Results

Why in DW?

Answer:

The basis of expression should be as stable as possible (that is the basis of expression should not vary also with the imposed conditions). Dry weight is in this way the most stable basis for expression (that has been largely used by our and another laboratories).

Conclusions

Minimize the conclusions

Answer:

Although we agree that this section is somewhat long, that is a consequence of our findings (what justifies that this section could be somewhat longer than usual). Still, to comply with this suggestion we made an effort to reduce the conclusion section.

Reviewer 2 Report

Introduction

Lane 64 to 87: Reduce to one paragraph and focus on what is stated in paragraph 3 (line 88 to 103) which is the most important idea

Materials and Methods

 247 - It was a denaturating agarose gel??

250- I consider that this step is not necessary, you could check the amplification of a housekeeping gene in a qPCR.

Results:

Table 2 - Move to supplementary data. This information is very useful and clear but it is more clearly read in graphs. I recommend making a figure
Figure 1 - It recommend to have an only figure with two graph, one for SOD and another for CAT. Is a better way to compare the enzymes of each genotypes. Another option is making per enzyme a graph for aCO2 and one for eCO2.

Table 3- Same as Table 2, make a figure is more proper

Discussions

Figure 2 - It should be move to Results. I recommend using the same criteria that I exposed before in figure 1.

Author Response

We thank the reviewer for their comments and suggestions.

We introduced the suggested modifications, but when needed we provide an answer regarding some specific questions.

Introduction

Lane 64 to 87: Reduce to one paragraph and focus on what is stated in paragraph 3 (line 88 to 103) which is the most important idea

Answer:

The two first paragraphs of the Introduction provide very relevant (namely the role of [CO2] rise in temperature, both of which are here studied – CP2 vs Heat). However, we understand the reviewer suggestion, and  to comply with it we significantly reduced the first two paragraphs.

Materials and Methods

247 - It was a denaturating agarose gel??

Answer: No. This is just done to assure the integrity of the RNA and the absence of smears.

  1. 250- I consider that this step is not necessary, you could check the amplification of a housekeeping gene in a qPCR.

Answer: We agree; this is just an additional quality control step.

Results:

Table 2 - Move to supplementary data. This information is very useful and clear but it is more clearly read in graphs. I recommend making a figure

Answer:

We do not know if we fully understood this suggestion. This table shows one of the most central presented data and should not be removed to “supplementary data”. Furthermore, if it would be placed as “supplementary data” there was no point of repeat this results presentation as a figure (thus, duplicating the results presentation).

Although we can agree that figures can usually be a good way to show results (we also use it in this manuscript for the enzymes activity) that is not the case in large data sets. Also, tables are a more accurate (and then, clear) way to provide this large data sets, since values are shown and statistical indexes can be easily placed near such values (what is much more complicated in graphs when several treatments are involved – as it is the case). It should be noted that the transformation of this table (and also Table 3) into figures/graphs would immensely increase the presentation of the results. For instance. just to compare, Fig 2 contains two enzymes presented in 6 graphs. That means that for the data of Table 2 we would need 6 new figures like Fig 2 (with a total of 36 graphs!). For table 3 (with the presentation per gene) another 6 Figures would be needed. At the end, the manuscript will have more than 80 graphs, what is not a clear way of presentation. We believe that such a large number of Figures is not a suitable way for presentation of this kind of large data sets, and that would make much more difficult to look and compare the obtained trends of different molecules/genes at each time/environmental condition. Therefore, having in mind that the scientific information is the same, we prefer to maintain the presentation in tables as a more accurate (and easy) way to provide the information for the readers.

Figure 1 - It recommend to have an only figure with two graph, one for SOD and another for CAT. Is a better way to compare the enzymes of each genotypes. Another option is making per enzyme a graph for aCO2 and one for eCO2.

Answer:

We did not fully understand the comments but we will try to answer, justifying our way of presentation of the results.

When it is said that is recommended to have “two graph, one for SOD and another for CAT” it means that we should discard the other enzymes (APX and GR)? If that is the case, we do not agree, for instance, due to the complementary action of APX and CAT in the H2O2 removal (as discussed in the paper) and to show that different enzymes from the ascorbate-glutathione antioxidative pathway respond differently.

The reviewer also says that “Is a better way to compare the enzymes of each genotypes”. However, we prefer to show each enzyme in the three genotypes to better understand if the pattern of response is similar or different among genotypes, although we performed an integrated discussion of the enzymes.

Regarding the suggestion “Another option is making per enzyme a graph for aCO2 and one for eCO2”. That would not be correct regarding the use of statistical indexes that are displayed for the mean comparison of temperature conditions (for each [CO2]) and between CO2 conditions (within each temperature). By separating the graphs in aCO2 and one for eCO2 such statistical comparison would be then performed between different graphs what does not seem a good choice.

Furthermore, all the Results presentation (and the Discussion) was built in this kind of reasoning. In our view, by following the suggestion of the reviewer would imply to totally re-write of these section(s), the total re-built of the graphs, all without any real advantage in such different presentation and because what can be concluded from these results would be the same.

Table 3- Same as Table 2, make a figure is more proper

Answer:

Please see the answer regarding Table 2, both due to the central importance of this data of gene expression (therefore they should not be sent to supplementary data) and the huge number of graphs needed to provide the exact same information (and without the same number accuracy or the information provided by the clear statistical indexes close to each value).

At this point we wish to underline that the reviewer do not present any question regarding the content (results) and just suggested alternative ways of presentations. Although we thank the reviewer for their suggestions, we decide not to follow them 1) for sake of simplicity and accuracy of the results presentation, 2) because a large number of graphs would worsen their readability, and 3) that important data, which is the core of this manuscript, should not be removed to supplementary data”.

Discussions

Figure 2 - It should be move to Results. I recommend using the same criteria that I exposed before in figure 1.

Answer:

The reviewer is correct when it says that Fig. 2 should be in the Results. Thanks for noticed that. We do not know why Plants place it in the Discussion section, but it was relocated now in the Results section.